# Health Related Values and Preferences Regarding Meat Intake: A Cross-Sectional Mixed-Methods Study

**DOI:** 10.3390/ijerph182111585

**Published:** 2021-11-04

**Authors:** Claudia Valli, Marilina Santero, Anna Prokop-Dorner, Victoria Howatt, Bradley C. Johnston, Joanna Zajac, Mi-Ah Han, Ana Pereira, Fernando Kenji Nampo, Gordon H. Guyatt, Malgorzata M. Bala, Pablo Alonso-Coello, Montserrat Rabassa

**Affiliations:** 1Department of Paediatrics, Obstetrics, Gynaecology and Preventive Medicine, Universidad Autónoma de Barcelona, 08193 Barcelona, Spain; marilinasantero@gmail.com; 2Iberoamerican Cochrane Centre, Biomedical Research Institute San Pau (IIB Sant Pau), 08025 Barcelona, Spain; PAlonso@santpau.cat (P.A.-C.); mrabassa.cochrane@gmail.com (M.R.); 3Department of Medical Sociology, Chair of Epidemiology and Preventive Medicine, Jagiellonian University Medical College, 31-034 Krakow, Poland; anna.prokop@uj.edu.pl; 4Faculty of Medicine, Dalhousie University, Halifax, NS B3H 4R2, Canada; vhowatt@dal.ca; 5Department of Community Health and Epidemiology, Dalhousie University, Halifax, NS B3H 4R2, Canada; 6Department Epidemiology and Biostatistics, School of Public Health, Texas A&M University, College Station, TX 77843, USA; bjohnston@dal.ca; 7Department of Nutrition, Texas A&M University, College Station, TX 77843, USA; 8Department of Hygiene and Dietetics, Chair Department of Epidemiology and Preventive Medicine, Jagiellonian University Medical College, 31-034 Krakow, Poland; joanna.faustyna.zajac@gmail.com (J.Z.); malgorzata.1.bala@uj.edu.pl (M.M.B.); 9College of Medicine, Chosun University, Gwangju 61452, Korea; mahan@chosun.ac.kr; 10Servicio Madrileño de Salud (SERMAS), 28008 Madrid, Spain; pereiraiglesiasana@gmail.com; 11Sociedad Madrileña de Medicina de Familia Comunitaria (SoMaMFyC), 28004 Madrid, Spain; 12Evidence-Based Public Health Research Group, Latin-American Institute of Life and Nature Sciences, Federal University of Latin-American Integration, Foz do Iguassu 85866-000, PR, Brazil; fernando.nampo@unila.edu.br; 13Department of Medicine, McMaster University, Hamilton, ON L8N 3Z5, Canada; guyatt@mcmaster.ca; 14CIBER de Epidemiología y Salud Pública (CIBERESP), 28029 Madrid, Spain

**Keywords:** health, values and preferences, red meat, processed meat, cross-sectional study, mixed methods, explanatory sequential, survey

## Abstract

Background. In addition to social and environmental determinants, people’s values and preferences determine daily food choices. This study evaluated adults’ values and preferences regarding unprocessed red meat (URM) and processed meat (PM) and their willingness to change their consumption in the face of possible undesirable health consequences. Methods. A cross-sectional mixed-methods study including a quantitative assessment through an online survey, a qualitative inquiry through semi-structured interviews, and a follow-up assessment through a telephone survey. We performed descriptive statistics, logistic regressions, and thematic analysis. Results. Of 304 participants, over 75% were unwilling to stop their consumption of either URM or PM, and of those unwilling to stop, over 80% were also unwilling to reduce. Men were less likely to stop meat intake than women (odds ratios < 0.4). From the semi-structured interviews, we identified three main themes: the social and/or family context of meat consumption, health- and non-health-related concerns about meat, and uncertainty of the evidence. At three months, 63% of participants reported no changes in meat intake. Conclusions. When informed about the cancer incidence and mortality risks of meat consumption, most respondents would not reduce their intake. Public health and clinical nutrition guidelines should ensure that their recommendations are consistent with population values and preferences.

## 1. Introduction

Many believe that people’s dietary choices have important consequences for their health. All individuals face the daily choice regarding what to eat, and in what quantity [1]. People’s food choices, in addition to social and environmental determinants, may depend on their beliefs regarding health effects, their beliefs about the environmental effects of their diet, the pleasure they take in eating, their social and cultural milieu and the relative importance they place on these issues.

When developing guidance for public dietary behaviour, respect for individual autonomy requires understanding the health-outcome-related values and preferences that are linked to diet among members of the public. Most dietary guidelines have, however, not only failed to conduct systematic reviews (SRs) of people’s values and preferences, but have also neglected this issue when making their recommendations [2,3].

With regard to meat, given the association between unprocessed red meat (URM) and processed meat (PM) consumption and adverse health outcomes (cancer and cardiovascular events) [4], dietary guidelines have generally recommended limiting meat intake [5,6,7]. In developing a guideline regarding meat consumption, our group undertook a SR that addressed relevant health-related values and preferences. We found that reasons for meat consumption varied and that people’s willingness to change their meat consumption is generally low [8], but because researchers had never undertaken the most relevant studies to inform the question, the evidence was only low quality.

We therefore developed and conducted a cross-sectional explanatory sequential mixed-methods study in order to evaluate adults’ values and preferences regarding URM and PM intake and their willingness to change their intake in the face of possible undesirable health consequences based on the dose–response meta-analysis SR of meat and cancer risk [9]. Unprocessed red meat was defined as mammalian meat (e.g., beef, pork, lamb), and processed meat was defined as white or red meat that was preserved by smoking, curing, salting, or by the addition of preservatives (e.g., hot dogs, charcuterie, sausage, ham, and cold cut deli meats). One serving corresponded to 120 g for unprocessed red meat, and 50 g for processed meat [10].

## 2. Methods

### 2.1. Study Design and Setting

This cross-sectional explanatory sequential mixed-methods study included a quantitative assessment through an online survey, a qualitative inquiry through semi-structured interviews, and a follow-up assessment through a telephone survey. Our team conducted the study in Spain between November 2020 and March 2021, based on a previously published study protocol where further details on the methods are provided [11]. The report follows STROBE guidelines [12].

This work constitutes one part of NutriRECS (Nutritional Recommendations; www.nutrirecs.com, accessed on 26 November 2020), an initiative that aims, by following a rigorous and transparent approach based on the methods promoted by the National Academy of Medicine, Guideline International Network and GRADE, and that includes the incorporation of values and preferences of the public [13], in order to develop trustworthy nutritional recommendations.

### 2.2. Study Population

People learned about this study thorough the Cochrane website and Twitter, where we published all of the information related to the study, eligibility criteria, contact information of the researcher carrying out the study, and the related link to access the online survey. People who were interested in participating completed the online consent form and accessed the survey. Respondents included adults between 18 and 80 years of age who currently consume URM and/or PM. We excluded adults who had active cancer and those who had suffered a major cardiovascular event such as: stroke, angina, myocardial infarction, heart failure, symptomatic peripheral arterial disease, as well as pregnant women and those unwilling or unable to provide informed consent.

### 2.3. Questionnaire and Study Procedures

The questionnaire was first developed and reviewed by experts on the topic in order to ensure the validity of the included items in the questionnaire; secondly, we pilot-tested it in English in a convenience sample of participants [14]. On the basis of the pilot study, our team modified the questionnaire, performed a translation into Spanish—one researcher translated the survey that was reviewed and a second researcher confirmed it—and finally, we developed an online version that we tested on 34 Spanish participants to establish clarity and understanding. Based on the findings of the pre-testing, we refined the survey to improve face and content validity. See Appendix A for the Spanish version of the online survey.

The questionnaire addressed the participants’ demographic characteristics, their medical history and meat consumption beliefs and behaviours, and it also included a direct-choice exercise. This exercise presented scenarios that were tailored to each individual’s weekly meat consumption and included, based on a prior SR and dose–response meta-analysis, the best estimates of the risk reduction of overall lifetime cancer incidences and cancer mortality that is associated with a decrease in URM and/or PM consumption [9]. In order to keep the presentation understandable and assimilable, we decided to focus only on cancer and thus, we omitted the possible cardiovascular effects. The scenarios also presented the corresponding certainty of the evidence for the potential risk reductions. The questionnaire was tailored to participants’ individual meat consumption (i.e., after they had stated their mean consumption, subsequent questions referred to those prior responses) and participants’ willingness to change their meat intake (those unwilling to change responded to additional questions regarding whether higher quality evidence or a larger effect would change their willingness).

Participants first considered the cancer-incidence scenario and expressed their willingness to “stop” their URM and PM intake using a 7-point Likert-scale with 1 (meaning definitely unwilling) and 7 (meaning definitely willing) (Question 1). If participants were unwilling to stop (≤4 of the Likert-scale), they were asked, using a 7-point Likert-scale question (Question a), if they would stop their intake if the certainty of the evidence was higher. If they were still unwilling to stop (≤4 of the Likert-scale), we asked them, using a multiple-choice question (Question b), if they were willing to stop if the evidence showed a larger risk reduction. If, after the above questions, participants were still unwilling to stop, we presented them with an additional 7-point Likert-scale question about their willingness to “reduce” their intake (Question 2). Similar to what was reported above, participants unwilling to reduce their intake (≤4 of the Likert-scale), were presented with the questions about the certainty of the evidence (Question a) and, if still unwilling, the magnitude of the risk (Question b). If participants were also unwilling to reduce their intake (≤4 of the Likert-scale), they were finally presented with a question about whether they were instead willing to increase their meat consumption using a 7-point Likert-scale question (Question 3). This logic of questions was applied for both types of meat and for both the cancer-incidence and cancer-mortality scenarios (Figure 1).

Two additional questions invited respondents to participate in a semi-structured interview and a follow-up assessment at 3 months. If the respondents had agreed to participate in the semi-structured interviews, then we arranged a meeting (through a secured Skype/Zoom call or by telephone) in which we reviewed and discussed their answers from the online survey and asked additional questions addressing their motives to change or continue with their current URM and/or PM consumption. At 3 months after the online survey, we conducted follow-up interviews via email and/or phone and asked the participants who had agreed to be contacted if they had made any changes in their meat consumption.

### 2.4. Data Synthesis and Analysis

#### 2.4.1. Quantitative Analysis

All statistical analyses were performed using RStudio (version 1.2.5033) [15]. Data were checked for normal distribution using the Kolmogorov–Smirnov test. An independent samples t-test (normal distribution) or a Mann–Whitney U test (non-normal distribution) was used to assess the differences between the two groups. For categorical variables, differences between groups were analyzed by the chi-square test. Missing values were excluded from the analysis.

We described the participants’ demographic and medical history information as well as their meat consumption behaviours using mean ± standard deviation or as median and inter-quartile-range (IQR) and number (percentage). Because the data were not normally distributed, we presented the participants’ willingness to stop, reduce and increase meat consumption in the face of undesirable cancer as medians and IQRs.

We performed a separate logistic regression analysis for each dependent variable in order to explore the determinants of the participants’ willingness to change meat consumption in the direct-choice scenarios. The dependent variables were the choice (unwilling versus willing) to stop and reduce eating URM and/or PM in the face of cancer-incidence risks as well as cancer-mortality risks. The team identified the independent variables of sex, age, level of education, occupational status and religious belief a priori as known potential confounders and they were included in each statistical model. Linear regression was not performed as planned in the protocol because the assumption of linearity was violated.

We calculated the number and percentage of participants who had made any changes in their meat consumption at the follow-up after three months.

#### 2.4.2. Qualitative Analysis

After collecting the data and transcribing the semi-structured interviews, we conducted an iterative, thematic analysis, using constant comparison within and across the transcripts of the study’s participants by following a six-step approach (i.e., familiarisation with the data, generating initial codes, searching for themes, reviewing the themes, defining and naming the themes and producing the final report) [16].

#### 2.4.3. Integrating Qualitative and Quantitative Analyses

We conducted a sequential analysis of the quantitative and qualitative components of the data. We analysed each dataset separately and then, at the end of the study, listed the findings from each component of our study and drew meta-inferences. Findings of interest from both data sets were compared and contrasted for convergence (whether findings from each data set agree), complementarity (whether findings offer complementary information on the same issue), dissonance (appear to contradict each other) and “silence” (a particular finding could only be explored in one data set) [17]. The integrated data were presented using a joint display [18], which presents each theme from the qualitative analyses according to the proportion that was obtained from the relevant online survey questions.

## 3. Results

### 3.1. Online Survey

#### 3.1.1. Participants’ Characteristics

Of the 304 individuals who participated in our study, typical respondents were women around 40 years old with a university degree (85%), employed (81%), and having at least one comorbidity (74%) (Table 1).

#### 3.1.2. Participants’ Meat Consumption Behaviour

Many participants reported consuming less than three servings of meat per week (76% of URM and 57% of PM), 24% of participants consumed three or more servings of URM and 43% of PM. Figure 2 presents the meat-consumption frequency behaviour. The type of URM most frequently consumed was beef or veal (76.0%) and, for PM, Serrano ham or shoulder ham (71.4%) (See Appendix A). The three main reasons for meat consumption among the participants included flavour, cost and availability, and were similar for URM and PM (See Appendix A).

With regard to URM consumption, 27.3% had previously reduced consumption for health; for PM, the same was true of 38.2% of participants, whereas 38.5% reported to have reduced their intake of meat in general for other non-health-related reasons. Among the eight different non-health-related reasons participants could choose from, animal welfare and environmental concerns were the most frequently reported (Table 2).

Unprocessed red meat was defined as mammalian meat (e.g., beef, pork, lamb), and processed meat was defined as white or red meat that was preserved by smoking, curing, salting, or by the addition of preservatives (e.g., hot dogs, charcuterie, sausage, ham, and cold cut deli meats). One serving corresponded to 120 g for unprocessed red meat and 50 g for processed meat.

#### 3.1.3. Willingness to Change Meat Consumption (Questions 1, 2 and 3)

The majority of participants were unwilling to introduce any changes to their URM and PM consumption in the face of the associated reductions in overall cancer-incidence and cancer-mortality risks. Most respondents were unwilling to stop their intake (URM: 78.6%; PM: 77.9%); of those unwilling to stop, most were also unwilling to reduce (URM: 81.1%; PM: 91.5%) their intake when presented with the cancer-incidence scenario; likewise, most participants were unwilling to stop (URM: 75.4%; PM: 76.4%), and of those unwilling to stop, to reduce (URM: 85.7%; PM: 80%) when presented with the mortality scenario. Similarly, none of the participants were willing to increase their URM and/or PM intake. Table 3 presents the participants’ willingness to stop, and if unwilling to stop, to reduce, and if unwilling to reduce, to increase URM and PM consumption in the face of cancer-incidence and cancer-mortality risks.

#### 3.1.4. Willingness to Change Meat Consumption with Higher Certainty (Questions a)

The availability of higher-certainty evidence affected the participants’ willingness to change their consumption in a minority of respondents who were unwilling to stop or reduce in response to the initial evidence presentation: 26.6% participants were willing to stop and 6.7% were willing to reduce their URM intake when they were presented with the cancer-incidence scenario. Similarly, with the cancer-mortality scenario, 19.0% were willing to sop and 6.9% were willing to reduce their intake. For PM, 35.8% of participants were willing to stop and 10.3% to reduce their intake when presented with the cancer-incidence scenario; similarly, for the cancer-mortality scenario, 29.2% were willing to stop and 13.6% to reduce. Table 3 presents the participants’ willingness to stop and reduce URM and PC consumption in the face of cancer-incidence and cancer-mortality risks with higher certainty.

#### 3.1.5. Willingness to Change Meat Consumption with a Larger Risk Reduction (Questions b)

The availability of a hypothetically larger reduction in cancer risk affected the willingness to change the meat consumption of some participants who were unwilling to stop or reduce in response to higher-certainty evidence: 37.0% participants reported to be willing to stop and 56.0% to reduce their URM intake when presented with the cancer-incidence scenario. Similarly, with the cancer-mortality scenario, 42.0% participants reported to be willing to stop and 54.0% to reduce their URM intake. For PM, 38.0% of participants were willing to stop and 50.0% to reduce their PM intake when presented with the cancer-incidence scenario, whereas in the cancer-mortality scenario, 55.0% of participants reported to be willing to stop and 53.0% to reduce their PM intake. Table 3 presents the participants’ willingness to stop and reduce URM and PC consumption in the face of cancer-incidence and cancer-mortality risks with a larger risk reduction.

#### 3.1.6. Predictors of Willingness to Change Meat Consumption

In the logistic regression analysis, gender appeared to be the only significant predictor of willingness to stop PM consumption in the cancer-incidence scenario (OR: 0.40; 95% CI: 0.15–0.93) and URM consumption in the cancer-mortality scenario (OR: 0.34; 95% CI: 0.11–0.88), with men being less willing to stop compared to women. Men also appeared to be less willing to stop eating PM (OR: 0.43; 95% CI: 0.18–0.96) and URM (OR: 0.27; 95% CI: 0.08–0.74) if the certainty was higher when presented with the cancer-incidence and cancer-mortality scenarios, respectively. Age, level of education, occupational status and religious belief did not appear to be significant predictors for any other dependent variables of willingness.

### 3.2. Semi-Structured Interviews

#### 3.2.1. Participants’ Characteristics

Of the 304 participants, seven agreed to participate in the semi-structured interviews; there were four men and three women, with a mean age of 38.6 years (SD = 5.0). All participants (100%) reported having a university degree, being employed, and six (86%) reported not having any comorbidity. Appendix A presents the participants’ sociodemographic and medical history.

#### 3.2.2. Participants’ Meat Consumption Behaviour

Participants’ meat consumption varied. Three participants consumed between 3 and 4 servings of PM per week, one participant consumed between 11 and 12 servings per week and three participants declared consuming less than one serving per week. Regarding URM, three participants declared to consume less than one serving per week, two declared consuming between 1 and 2 servings per week and two consumed between 3 and 4 servings per week (See Appendix A).

When asked if they had reduced their meat consumption in the past for health reasons and/or for other reasons, three participants declared having reduced both their URM and PM intake in the past due to health reasons, two participants reported having reduced their intake for animal welfare and environmental concerns and one participant reported cost as the main reason for having reduced his consumption. From the survey, none of the participants reported to be willing to stop or reduce their meat intake in the future.

#### 3.2.3. Meat Consumption Preferences

We have identified three main themes reflecting the participants’ preferences: (1) Social and/or family context of meat consumption, (2) Health- and non-health-related concerns about meat, and (3) Uncertainty of the evidence. Here we present some quotations from research participants.

##### Social and/or Family Context Meat Consumption

Two participants did not consider themselves regular meat eaters and reported eating meat mainly in social contexts.

“I’m not vegetarian and not vegan either, but if it was for me, I wouldn’t choose meat as part of my daily meals. But once in a while if I go out with friends, I do eat it. I haven’t eaten meat on a regular basis for a year now” (Female participant, 33 years old)

“I have not eaten meat on a regular basis for many years now. I consume meat especially for social occasions” (Male participant, 41 years old)

One participant reported consuming meat for its nutritional properties and mainly in social contexts.

“I have not completely stopped eating meat, as I consider it necessary to have certain nutritional values such as iron or vitamin B12. In addition, due to my origin one of my favourite foods is Iberian ham. On the other hand, the meat that I usually consume is of high quality and does not usually come from large farms. Even, for tradition, I consume game meat when I return to the family home” (Male participant, 32 years old)

One participant reported consuming meat mainly for the health and nutritional needs of her family.

“If it was for me, I would follow a more vegetarian diet, but I have to adapt to the needs of my children and family” (Female participant, 39 years old)

##### Health- and Non-Health-Related Concerns about Meat

Two participants reported health as the main reason for having reduced their meat intake in the past.

“In 2015 when I became a mother, I started to look for information about nutrition and get more information about what was healthy to take care of me and my son, that is when I decided to reduce my meat consumption” (Female participant, 39 years old)

“I had this idea that meat was high in fat and more expensive. So, I started to reduce my meat consumption, especially red meat, and in the end, I was eating mostly chicken. Gradually, I started to remove all types of meat from my daily meals” (Female participant, 33 years old)

Two participants highlighted other aspects that should be of concern when consuming meat. Animal welfare and/or environmental concerns were stated as important aspects to be considered when consuming meat.

“In recent years, there has been a lot of investigative journalism about the situation of large-scale animal farms and the deplorable conditions in which they are raised. In addition, livestock farming is directly related to greenhouse gas emissions and the deforestation of huge regions to grow pasture and feed for livestock. Livestock farming is one of the human activities that generates the most CO_2_ emissions” (Male participant, 32 years old)

“From what I have read, too much meat can lead to diseases but on the other hand I am concerned about the sustainability aspects related to its consumption. This doesn’t mean I don’t eat meat, but I don’t buy processed meat. I do eat beef sometimes and when I buy it, I go to the butcher so that I can choose the type of meat, the cut, and make sure of the origin” (Male participant, 41 years old)

##### Uncertainty of the Evidence

Three participants reported that the certainty of the evidence was not sufficiently convincing to cause changes in their meat consumption.

“I have no proof, nor enough evidence to think that I should reduce my consumption. If the evidence said that there was a real and significant reduction, I would reduce my consumption.” (Male participant, 39 years old).

“I like meat, and it is for sure a barrier to reduce or quit its consumption, especially when the evidence is unclear.” (Male participant, 47 years old).

“As far as I can see, the evidence is not valid enough to completely stop eating meat.” (Female participant, 39 years old).

### 3.3. Integrated Data

In Table 4, the data from the quantitative (online survey) and qualitative (semi-structured interviews) analyses are integrated and presented in a joint display, which allows a deeper understanding of the participants’ values and preferences around meat consumption. The quotes from the transcripts that most clearly represent the participants’ views have been included in the right column. Table 4 will be interpreted in the discussion.

### 3.4. Follow-Up Assessment at 3 Months

The same seven participants who participated in the semi-structured interviews completed the follow-up assessment, with the addition of one woman participant; four men and four women with a mean age of 39.3 years (SD = 5.0) participated. Five participants (63%, three men and two women) reported not having made any changes in their URM and PM consumption, two participants (25%, one man and one woman) reported having increased their meat intake—one participant for URM and the other for PM—and finally, one woman participant (12%) reported having reduced the intake of PM.

## 4. Discussion

### 4.1. Main Findings

In this cross-sectional explanatory sequential mixed-methods study that included more than 300 adults in Spain, we found that, in the face of the available evidence regarding cancer-incidence and cancer-mortality risk reductions they would achieve, most people were unwilling to reduce their meat intake. Men were appreciably less willing to reduce meat consumption than were women. In the semi-structured interviews, participants reported consuming meat in social contexts and/or in response to family preferences. Health proved to be one important factor in favour of consuming meat and other aspects such as environmental concerns emerged as important considerations. Three of seven participants reported that the evidence was too uncertain for them to make changes in their current consumption. Overall, quantitative and qualitative findings were in agreement.

The included participants can be considered as infrequent meat eaters since the majority consumed between 1 and 2 servings of meat per week versus the estimated average consumption of three servings of meat per week [19]. This could explain why people who already had a low meat consumption were not willing to further decrease their meat intake. In fact, during the semi-structured interviews, some participants did not consider themselves as regular meat eaters and reported consuming meat occasionally, mainly in social contexts or because of tradition and/or family preferences. The participants’ unwillingness to reduce or increase consumption suggests that participants were satisfied with their meat consumption habits and did not feel the need to make any changes; as emerged during the interviews, people felt that they were already consuming a healthy amount of meat that did not need to be changed.

### 4.2. Our Results in the Context of Previous Research

Our results are similar to the findings from a previous mixed-methods systematic review that was conducted by our team [8]. In this review, we showed that most omnivores were unwilling to change their meat intake. More recent studies also show a low willingness to change meat consumption [14,20,21]. Both our review and further studies also showed that men were more attached to meat consumption, and less willing to change their intake. In addition, although our results showed that participants were unwilling to reduce their meat in the face of cancer risks, many had reduced their intake in the past for other aspects, such as environmental concerns and animal welfare reasons. These aspects, which emerged during the interviews, are similar to the conclusions of a recent systematic review that found that environmental motives were already appealing to significant proportions of Western meat-eaters, who were adopting certain meat-curtailment strategies such as meat-free days [22].

### 4.3. Strengths and Limitations

Our study has several strengths. It is the first study, to our knowledge, that has comprehensively and explicitly evaluated people’s health-related values and preferences, and their willingness to change meat consumption when informed of the potential adverse cancer risk and the uncertainty around this evidence. The information that patients received was based on a recent rigorous dose–response meta-analysis [9]. We developed and published a protocol reporting this study’s methodology [13]. We followed an explanatory mixed-methods approach to the collection of both quantitative and qualitative evidence that enhanced the interpretability of our results. We used health states to ensure a similar understanding among participants of the presented outcomes.

Our study also has some limitations. Most of the included participants had a university degree and consumed less than three servings per week, which was the average meat intake in Spain [19]; therefore, our results might not be representative for the rest of the Spanish population. Although we provided information about the associated reductions in cancer risk in different formats, we did not check for understanding. We also only presented data on cancer risk and did not present other health risks, such as cardiovascular effects, in order not to overburden the participants. In addition, while the semi-structured interviews and follow-up assessment findings were collected from a small proportion and convenience sample of participants (only 7 and 8 participants agreed to participate, respectively); however, their sociodemographic characteristics and their meat consumption behaviours were very similar to the rest of study’s participants. The response rate for the survey questions on willingness varied. The less willing they were to change meat consumption, the more questions a participant had to answer (see study procedures).

### 4.4. Implications for Practice and Research

This study will be informative in the development of both public health and clinical nutritional recommendations regarding meat consumption. For example, given that people are unlikely to modify their meat consumption on the basis of small and uncertain health benefits, panels would be more likely to make conditional rather than strong recommendations for the reduction of meat consumption for healthcare reasons. Our study provides guidance on the methods and procedures of how to conduct an exploratory sequential mixed-methods observational study that aims to identify people’s health-related values and preferences. Future research is needed to replicate this study in other populations with higher meat intake and in other settings and cultures. The design we used could be applicable to other foods and/or nutrients, settings and/or nutritional contexts.

## 5. Conclusions

When informed about the cancer incidence and mortality risks of meat consumption, most respondents would not reduce their intake. Organizations developing public health and clinical nutrition guidelines should ensure their recommendations are consistent with population values and preferences.

## Figures and Tables

**Figure 1 ijerph-18-11585-f001:**
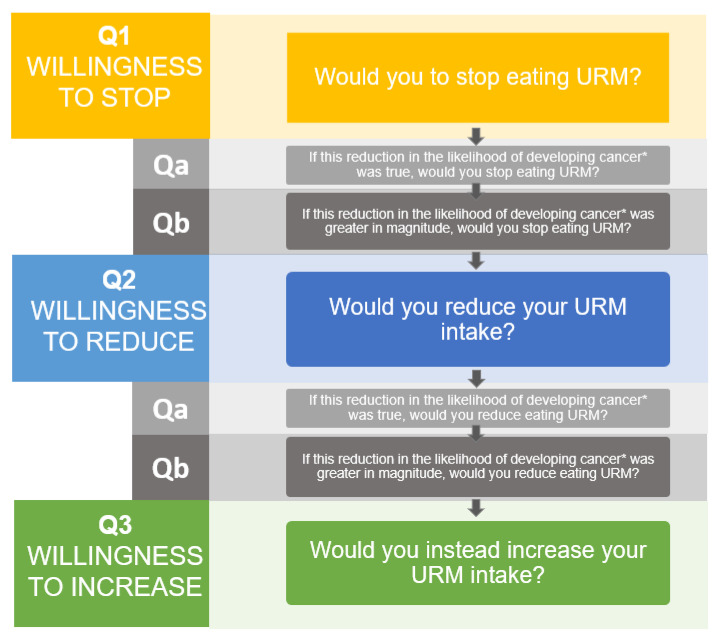
Questions framework for the direct-choice cancer-incidence exercise for unprocessed red meat. Abbreviations: URM = unprocessed red meat; Q1 = Question 1; Q2 = question2; Q3 = Question3; Qa = Question a; Qb = Question b. Q1–Q2–Q3: Willingness to stop, reduce and increase meat intake was based on a 7-point Likert-scale with 1 (meaning definitely not) and 7 (meaning definitely yes). Qa: Willingness to stop and reduce meat intake with higher certainty was based on a 7-point Likert-scale with 1 (meaning definitely not) and 7 (meaning definitely yes). Qb: Willingness to stop and reduce processed meat consumption with a larger risk reduction was formulated as a multiple-choice question. This logic of questions was applied for both types of meat and for both cancer-incidence and cancer-mortality scenarios. * For the mortality scenarios “developing cancer” was changed into “dying from cancer”.

**Figure 2 ijerph-18-11585-f002:**
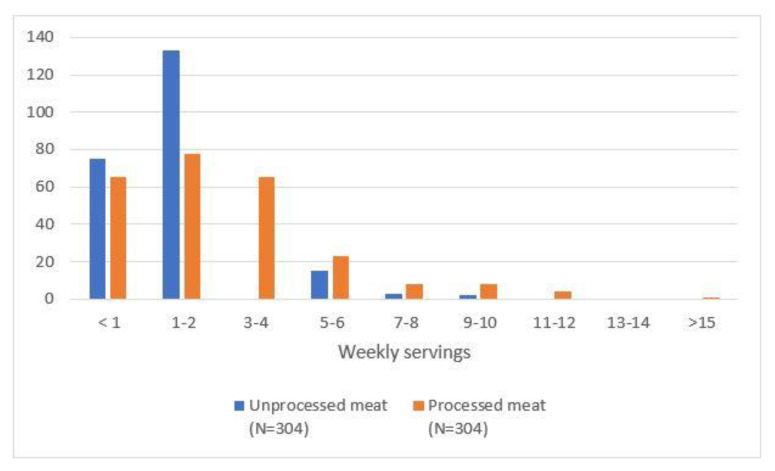
Meat consumption frequency behavior.

**Table 1 ijerph-18-11585-t001:** Participants’ sociodemographic and medical history.

	Overall (*n* = 304)
Sex, *n* (%)	
Women	189 (62.0)
Men	115 (38.0)
Age, years	
Mean (SD)	39.8 (10.7)
Median (Q1, Q3)	38.0 (32.0, 46.0)
Education level, *n* (%)	
Primary education	3 (1.0)
Secondary education	14 (4.6)
Professional education	24 (7.9)
University education	259 (85.2)
No studies	1 (0.3)
Employment status, *n* (%)	
Employed	247 (81.2)
Unemployed	34 (11.2)
Student	20 (6.6)
Marital status, *n* (%)	
Married	94 (30.9)
Common-law couple	5 (1.6)
Living with my partner or family	87 (28.6)
Separated	2 (0.7)
Divorced	12 (3.9)
Widow/widower	1 (0.3)
Single	100 (32.9)
Children, *n* (%)	
One child	42 (13.8)
Two children	62 (20.4)
Three or more children	14 (4.6)
None	183 (60.2)
Religion, *n* (%)	
Catholicism	62 (20.4)
Other	9 (3.0)
None	230 (75.7)
Physical activity intensity ^¥^, *n* (%)	
Low	82 (27.0)
Moderate	139 (45.7)
High	80 (26.3)
Weight, kg	
Mean (SD)	69.9 (14.5)
Median(Q1, Q3)	68.0(59.8, 79.0)
Height, m	
Mean (SD)	1.70 (0.1)
Median (Q1, Q3)	1.70 (1.6, 1.8)
BMI	
Mean (SD)	24.3 (4.1)
Median (Q1, Q3)	23.6 (21.5, 26.2)
Comorbidities, *n* (%)	
Hormonal system disorders	14 (4.6)
Digestive diseases	12 (3.9)
Musculoskeletal disorders	8 (2.6)
Other	41 (13.5)
None	226 (74.3)
Family history of cancer, *n* (%)	
Yes	198 (65.1)
No	73 (24.0)
I don’t know	30 (9.9)

Abbreviations: SD = standard deviation; Q1 = Quartile 1; Q3 = Quartile 3, kg = kilograms; m = meters; BMI = body mass index. ^¥^ Physical activity (PA) intensity was categorized as follows: participants who reported doing PA every day were categorized in the “high” category; who reported doing PA at least once a week was categorized in the “moderate” one and the rest of participants were categorized in the “low” category.

**Table 2 ijerph-18-11585-t002:** Participants’ meat reduction in the past.

Past reduction due to health reasons	
Unprocessed red meat	
N	283
No, *n* (%)	200 (65.8)
Yes, *n* (%)	83 (27.3)
Processed meat	
N	283
No, *n* (%)	167 (54.9)
Yes, *n* (%)	116 (38.2)
**Past reduction due to other reasons**	
Meat in general	
N	282
No, *n* (%)	165 (54.3)
Yes, *n* (%)	117 (38.5)
**Other reasons, *n* (%)**	
Animal welfare	62 (20.4)
Environmental concerns	67 (22.0)
Family preferences	15 (4.9)
Social context	7 (2.3)
Availability/accessibility	5 (1.6)
Flavour	21 (6.9)
Cost	14 (4.6)
Other	31 (10.2)

**Table 3 ijerph-18-11585-t003:** Willingness to change meat consumption in the face of cancer-incidence and cancer-mortality risks.

	URM	PM
**Willingness to stop—Question 1**		
Cancer Incidence		
N	126	163
Willing, *n* (%)	27 (21.4)	36 (22.1)
Unwilling, *n* (%)	99 (78.6)	127 (77.9)
Median	3.0	3.0
Q1, Q3	(1.0, 4.0)	(2.0, 4.0)
Cancer Mortality		
N	118	157
Willing, *n* (%)	29 (24.6)	37 (23.6)
Unwilling, *n* (%)	89 (75.4)	120 (76.4)
Median	3.0	3.0
Q1, Q3	(1.0, 4.0)	(2.0, 4.0)
**Willingness to stop with higher certainty—Question a**		
Cancer Incidence		
N	94	120
Willing, *n* (%)	25 (26.6)	43 (35.8)
Unwilling, *n* (%)	69 (73.4)	77 (64.2)
Median	3.0	3.0
Q1, Q3	(2.0, 5.0)	(2.0, 5.0)
Cancer Mortality		
N	84	106
Willing, *n* (%)	16 (19.0)	31 (29.2)
Unwilling, *n* (%)	68 (81.0)	75 (70.8)
Median	3.0	3.0
Q1, Q3	(1.0, 4.0)	(2.0, 5.0)
**Willingness to stop with a larger risk reduction—Question b**		
Cancer Incidence		
N	68	50
Unwilling, *n* (%)	21 (31.0)	17 (34.0)
Willing, *n* (%)	25 (37.0)	19 (38.0)
Neither unwilling nor willing, *n* (%)	22 (32.0)	14 (28.0)
Cancer Mortality		
N	67	74
Unwilling, n (%)	21 (31.0)	17 (23.0)
Willing, n (%)	28 (42.0)	41 (55.0)
Neither unwilling nor willing, n (%)	18 (27.0)	16 (22.0)
**Willingness to reduce—Question 2**		
Cancer Incidence		
N	37	47
Willing, *n* (%)	7 (18.9)	4 (8.5)
Unwilling, *n* (%)	30 (81.1)	43 (91.5)
Median	3.0	2.0
Q1, Q3	(1.0, 4.0)	(1.0, 3.0)
Cancer Mortality		
N	35	30
Willing, *n* (%)	5 (14.3)	6 (20.0)
Unwilling, *n* (%)	30 (85.7)	24 (80.0)
Median	3.0	3.0
Q1, Q3	(1.0, 4.0)	(2.0,4.0)
**Willingness to reduce with higher certainty—Question a**		
Cancer Incidence		
N	30	39
Willing, *n* (%)	2 (6.7)	4 (10.3)
Unwilling, *n* (%)	28 (93.3)	35 (89.7)
Median	3.0	3.0
Q1, Q3	(1.0, 4.0)	(2.0, 4.0)
Cancer Mortality		
N	29	22
Willing, n (%)	2 (6.9)	3 (13.6)
Unwilling, n (%)	27 (93.1)	19 (86.4)
Median	2.0	3.0
Q1, Q3	(1.0, 4.0)	(1.3,4.0)
**Willingness to reduce with a larger risk reduction—Question b**		
Cancer Incidence		
N	27	20
Unwilling, *n* (%)	12 (44.0)	10 (50.0)
Willing, *n* (%)	15 (56.0)	10 (50.0)
Neither unwilling nor willing, *n* (%)	0 (0)	0 (0)
Cancer Mortality		
N	26	20
Unwilling, *n* (%)	12 (46.0)	10 (50.0)
Willing, *n* (%)	14 (54.0)	10 (50.0)
Neither unwilling nor willing, *n* (%)	0 (0)	0 (0)
**Willingness to increase—Question 3**		
Cancer Incidence		
N	22	25
Willing, *n* (%)	0	0 (0.0)
Unwilling, *n* (%)	22 (100.0)	25 (100.0)
Median	1.0	1.0
Q1, Q3	(1.0, 2.0)	(1.0, 2.0)
Cancer Mortality		
N	13	13
Willing, *n* (%)	0 (0.0)	0 (0.0)
Unwilling, *n* (%)	13 (100.0)	13 (100.0)
Median	1.0	1.0
Q1, Q3	(1.0, 1.0)	(1.0, 4.0)

Abbreviations: URM = unprocessed red meat, PM = processed meat, Q1 = Quartile 1; Q3 = Quartile 3. Question 1,2,3: Willingness to stop and reduce meat intake was based on a 7-point Likert-scale with 1 (meaning definitely not) and 7 (meaning definitely yes). Question a: Willingness to stop and reduce meat intake with higher certainty was based on a 7-point Likert-scale with 1 (meaning definitely not) and 7 (meaning definitely yes). Question b: Willingness to stop and reduce unprocessed red meat consumption with a larger risk reduction was formulated as a multiple-choice question. Unwilling = ≤4 of the Likert-scale, Willing = ≥ 5 of the Likert-scale. The sample size (N) varied across the Willingness and cancer scenarios and type of meat because the questionnaire was tailored according to the participants’ responses.

**Table 4 ijerph-18-11585-t004:** Joint display of integrated data from qualitative and quantitative data sets.

Qualitative Data	Quantitative Data		Representative Quotes	Interpretation
Semi-Structured Interview Themes	Online Survey Questions	Online Survey Results		
Social and/or family context meat consumption	What are the most important factors that favour your consumption of red meat and processed meat? Select all that apply *	Social context was selected as a factor favouring unprocessed red meat and processed meat consumption by 52% and 40% of participants respectively.	*“I consume meat especially social occasions”*	Participants reported that social gatherings influenced their meat consumption.
		Family preference was selected as a factor favouring unprocessed red meat and processed meat consumption by 50% and 33% of participants respectively.	*“I have to adapt to the needs of my children and family”*	Participants reported that family preference influenced their meat consumption.
		Tradition was selected as a factor favouring unprocessed red meat and processed meat consumption by 57% and 33% of participants respectively.	*“* *Even, for tradition, I consume game meat when I return to the family home”*	Participants reported that tradition influenced their meat consumption
Health- and non-health-related concerns about meat	What are the most important factors that favour your consumption of unprocessed red meat? Select all that apply *	Health was selected by 41% of participants as a factor favouring unprocessed red meat consumption.	*“I consider red meat necessary to have certain nutritional values such as iron or vitamin B12”*	Participants highlighted the nutritional value of unprocessed red meat as a reason for consuming it.
	In the past, have you cut back on red and/or processed meat for non-health reasons?	Environmental concerns were selected by 22% of participants.The second highest selected reason as a non-health-related reason for having reduced meat consumption in the past.	*“Livestock farming is one of the human activities that generates the most CO_2_ emissions”*	Non-health-related reasons such as environmental concerns play an important role in people’s meat consumption habits.
Uncertainty of the evidence	What are the most important factors that favour your consumption of unprocessed red meat and processed meat? Select all that apply *	Taste was selected as a factor favouring unprocessed red meat and processed meat consumption by 79% and 49% of participants respectevely. The most selected factor.	*“I like meat, and it is for sure a barrier to reduce or quit its consumption, especially when the evidence is unclear”*	Taste was one of the most voted factors for consuming meat, and this could explain why in the face of uncertain evidence, participants were unwilling to stop and/or reduce their intake.

* 11 factors were provided to choose from, see Appendix A.

## Data Availability

All data generated or analysed during this study are included in this published article and its Appendix A.

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
