# Peer review of "Health Related Values and Preferences Regarding Meat Intake: A Cross-Sectional Mixed-Methods Study"

_ijerph, 2021, doi:10.3390/ijerph182111585_

Round 1

Reviewer 1 Report

 I looked into your references and notice that there were no studies that examined social relations and those who support eaten meat versus who don't. At least one study found that some reduced their meat eating because their physicians or family members put pressure to eat less. 

I am not sure whether your measures of capures values. Seams to me that you have captured norms and beliefs.

Reviewer 2 Report

The current manuscript describes a qualitative approach to evaluate individuals' willingness to change health-related dietary habits. The suggestions below are aimed at improving the stand-alone nature and overall clarity of presentation of this manuscript.

  1. Line 74-78 - this section appears to detract from your stated Aims and could be placed elsewhere (e.g. Acknowledgements?) so as not to detract from the flow of written work.
  2. Methods - provide further detail on the sample size involved here or the overall approach to sampling. The total number would appear below that which might be assumed to be nationally representative at first glance.
  3. Methods - further detail into the choice of question content and questionnaire design would be useful to ensure that this manuscript stands alone from previous work. Can the authors explain further the approach to question structure and question design to help rationalise the approach undertaken.
  4. Discussion line 421-433 - the authors note that intake of red meat/red meat products was not representative of the wider Spanish population but have not supported this with nationally-representative data. Please support your statement with appropriate citations. 
  5. Section 4.4 (and possible Conclusions) - your findings highlight unwillingness of individuals to follow existing guidelines. However it's unclear how these findings could inform future iterations of public health and clinical nutritional recommendations as stated. Either further details needs to be added to explain this point further or the statement should be revised to better align with the focus of the current research. Are authors referring to public-health messaging (related to adoption of more notionally ideal dietary habit) as opposed to changing of recommendations themselves? Furthermore, it's unclear why the focus of the questions was on cancer versus other health issues associated with higher intake of red meat and its products.

Reviewer 3 Report

The manuscript aimed to evaluate adults’ values and preferences regarding unprocessed red meat (URM) and processed meat (PM) and their willingness to change their consumption in the face of possible undesirable health consequences”. The theme is interesting, and I have some recommendations to improve it before publication.

  • Abstract: “People’s values and preferences determine daily food choices” – there are several issues that determine daily food choices. Not only people’s values and preferences. Mention if you used a convenience sample. Describe your target audience.
  • Adjust the reference format in the text.
  • Line 91 – “Respondents included adults between 18 and 80 years” – Are all of them considered adults? Are people >60 considered elderly?
  • Line 97 – Remove “(Error! Reference source not found.)”. Insert the questionnaire reference.
  • Lines 98 – 100: Describe the questionnaire translation and modification process and the internal validation.
  • Insert the questionnaire as an Appendix.
  • How did you process the “blank responses”; did you lost participants during the research period?
  • Line 117 - Participants first considered the cancer incidence scenario and expressed their willingness to “stop” their URM and PM intake
  • Line 190 – exclude (Error! Reference source not found.) and insert the reference.
  • Adjust the title below the figures.
  • Lines 221 – 224 – This information should be placed in the first moment that you mentioned URM and PM in the manuscript.
  • Table 2 – explain the reduced number of responses compared to your total sample.
  • Line 255 – changhe “sop” to “stop”
  • Line 298 – Explain “AW and EC”
  • Line 395 – remove the “.”
  • Line 402 – remove (Error! Reference source not found.)
  • All the supplementary material should be mentioned in the main text.

Thank you for the opportunity to review this manuscript!

Round 2

Reviewer 1 Report

no more concerns

Reviewer 2 Report

Authors have expanded on all the points raised in an appropriate and effective manner.